# Synergy of Mutation-Induced Effects in Human Vitamin K Epoxide Reductase: Perspectives and Challenges for Allo-Network Modulator Design

**DOI:** 10.3390/ijms25042043

**Published:** 2024-02-07

**Authors:** Marina Botnari, Luba Tchertanov

**Affiliations:** Centre Borelli, École Normale Supérieure (ENS) Paris-Saclay, Centre National de la Recherche Scientifique (CNRS), Université Paris-Saclay, 4 Avenue des Sciences, F-91190 Gif-sur-Yvette, France; botnarimarina97@gmail.com

**Keywords:** hVKORC1, missense mutations, protein folding, intrinsic disorder, molecular recognition, allosteric regulation, protein–protein interactions, PDI–hVKORC1 complex, 3D modelling, molecular dynamics, computational biophysics

## Abstract

The human Vitamin K Epoxide Reductase Complex (hVKORC1), a key enzyme transforming vitamin K into the form necessary for blood clotting, requires for its activation the reducing equivalents delivered by its redox partner through thiol-disulfide exchange reactions. The luminal loop (L-loop) is the principal mediator of hVKORC1 activation, and it is a region frequently harbouring numerous missense mutations. Four L-loop hVKORC1 mutants, suggested in vitro as either resistant (A41S, H68Y) or completely inactive (S52W, W59R), were studied in the oxidised state by numerical approaches (in silico). The DYNASOME and POCKETOME of each mutant were characterised and compared to the native protein, recently described as a modular protein composed of the structurally stable transmembrane domain (TMD) and the intrinsically disordered L-loop, exhibiting quasi-independent dynamics. The DYNASOME of mutants revealed that L-loop missense point mutations impact not only its folding and dynamics, but also those of the TMD, highlighting a strong mutation-specific interdependence between these domains. Another consequence of the mutation-induced effects manifests in the global changes (geometric, topological, and probabilistic) of the newly detected cryptic pockets and the alternation of the recognition properties of the L-loop with its redox protein. Based on our results, we postulate that (i) intra-protein allosteric regulation and (ii) the inherent allosteric regulation and cryptic pockets of each mutant depend on its DYNASOME; and (iii) the recognition of the redox protein by hVKORC1 (INTERACTOME) depend on their DYNASOME. This multifaceted description of proteins produces “omics” data sets, crucial for understanding the physiological processes of proteins and the pathologies caused by alteration of the protein properties at various “omics” levels. Additionally, such characterisation opens novel perspectives for the development of “allo-network drugs” essential for the treatment of blood disorders.

## 1. Introduction

The human vitamin K epoxide reductase (hVKORC1), an endoplasmic reticulum (ER) protein, reduces inactive vitamin K 2,3-epoxide to active vitamin K quinone, required for blood coagulation [1,2,3,4]. Four functional cysteine residues of hVKORC1 control all steps of its repeatedly reproduced enzymatic cycle. Two cysteine residues, C^43^ and C^51^, are located in the ER luminal loop (L-loop), and two others, C^132^ and C^135^, are positioned in the luminal end of the transmembrane domain (TMD), forming the C^132^XXC^135^ motif of the active site. To regain the catalytic activity of hVKORC1, these cysteines must be reduced by an external redox protein through a C^43^- and C^51^-mediated sequential electron-transfer process. This process is required prior to intra-protein electron transfer to the C^132^XXC^135^ motif involved in the vitamin K transformation. Using numerical modelling, we suggested that the protein disulfide isomerase (PDI) is the most probable hVKORC1 redox partner, initiating its activation by delivering the reducing equivalents [5]. Our in silico prediction was later confirmed by in vitro data [6].

hVKORC1 is a small (163 amino acids) modular and multifaceted protein consisting of structurally diverse domains located in different environments: a well-folded TMD placed in the membrane, an intrinsically disordered L-loop protruding into the ER, and highly flexible N- and C-terminals floating in the cytoplasm. Each of these domains fulfils its specific role, complementing and ensuring the functions of other regions of hVKORC1, providing the global implementation of mutually interconnected and tightly regulated functions of hVKORC1 [5,7].

The perfectly controlled multi-step enzymatic process of the native hVKORC1 is frequently deregulated by missense mutations. A genetic polymorphism of hVKORC1 associates with low or accelerated vitamin K recycling rates [4,8,9,10], causing serious diseases such as haemorrhages and thrombosis, including enhanced thrombogenicity in severe COVID-19 cases [11]. Moreover, the hVKORC1 polymorphism affects the anti-vitamin K anticoagulant (AVK) drug dose responses, promoting resistance to treatment [10,12,13]. Currently available AVKs are the vitamin K-competitive coumarin or indanedione derivatives [14]. If hVKORC1 mutations are frequently associated with resistance phenomena, the hVKORC1 physiological response to AVKs is also highly patient-dependent. In vitro studies of 25 hVKORC1 mutants showed that only six increased hVKORC1 resistance to AVKs, and others led to a loss of activity [8]. These mutations are located either in the TMD or in the L-loop. Curiously, out of 45 L-loop residues (R33-N77), 19 (more than 40%) show different genetic variations.

Therefore, the study of hVKORC1 and its mutants is of significant interest for fundamental research to understand their activation by the protein redox through thiol-disulfide exchange reactions [15,16,17], a crucial process in biology that plays a primary role in protective mechanisms against oxidative stress or redox regulation of cell signalling [18]. The study of these proteins, which constitute clinically relevant targets, is also pursued, with applied objectives focused both on the understanding of the mechanisms of mutation-induced resistance to AVKs and on the development of strategies aimed at modulating its aberrant activity by alternative approaches, other than classical inhibition of the active site.

Such research can be based on the pioneering three-dimensional (3D) de novo model of hVKORC1, which established the four-helices hVKORC1 topology of TMD [19] (Figure 1A). The correctness of this model was later confirmed by crystallographic studies [20]. This model was used to generate putative functionally related enzymatic states of hVKORC1, which were further confirmed by assessing their capacity to recognise and bind vitamin K and AVKs. In silico-predicted binding energies were found to be highly correlated with experimental inhibiting constants [19]. Moreover, this de novo model of hVKORC1 and the crystallographic structure of the redox protein PDI (in oxidised and reduced states, respectively) were carefully studied by in silico methods and used to generate the first molecular PDI/hVKOC1 precursor complex [5,7].

In this work, we concentrate on four L-loop mutants possessing the point missense mutations: two (A41S and H68Y) associated with the resistance phenotype observed in patients, and two others (S52W and W59R) leading to a loss of hVKORC1 activity, as was reported by in vitro determination of kinetic and inhibition constants [8]. Our choice of mutants was not influenced by the level of their responses to AVKs or vitamin K, but guided by differences in physico-chemical properties between native and mutated residues as well as their position on the L-loop (Figure 1B).

The point mutations A41S and S52W are located in the “cap” region, stabilised by a disulfide bridge S-S formed by C^43^ and C^51^, groups directly involved in interconversion with the dithiol of the redox Protein Disulfide Isomerase (PDI). The remaining W59R and H68Y are positioned in the “hinge”, a fully relaxed L-loop region, involved in PDI recognition [5].

We do not explore the recognition of hVKORC1 mutants by AVKs targeting the reduced (active) state of the enzyme. Our primary focus lies in identification of the effects caused by mutations on the oxidised (inactive) state of hVKORC1, which is a target of its redox protein PDI. Using purely numerical approaches such as 3D modelling and molecular dynamics (MD) simulation, we investigated mutation-induced effects on the structure and dynamics of hVKORC1 in the fully oxidised state. We compared the inherent hVKORC1 mutation-induced effects and interpreted them in structural, dynamical, and free energy-related terms, completing the DYNASOME [21] description of each mutant. Next, we explored the intra-protein pockets in the native hVKORC1 and its mutants to describe the POCKETOME [22] of each protein studied. DYNASONE and POCKETOME constitute the primary basis for both objectives: deciphering the fundamental mechanisms regulating the abnormal hVKORC1 mutants’ functions (aberrant activation, resistance phenomenon, atypical enzymatic transformation, etc.) and their practical application in the clinic and pharmacology.

Given that the active site of hVKORC1 has traditionally been used for AVK development, we paid special attention to the L-loop, suggesting its potential use as an alternative pharmacological target. We hypothesised that identifying the L-loop allosteric pockets would allow its use as an innovative target for the development of novel drugs, non-competitive with vitamin K, which would inhibit the L-loop plasticity required for hVKORC1 binding to its redox protein.

Despite the clear advantage of allosteric drugs, their design in many cases does not always succeed [23,24,25]. Therefore, we explored the refined models of hVKORC1^A41S^, hVKORC1^H68Y^, hVKORC1^S52W^, and hVKORC1^W59R^ mutants (Figure 1C) as putative PDI targets, similarly to hVKORC1^WT^ [5,7]. Identification of hVKORC1 interactions with PDI may deliver another therapeutic target—the interaction interface formed during the recognition and multistep binding processes of PDI to the L-loop. These interaction interface modulators may regulate or prevent aberrant hVKORC1 activation.

The use of these two types of protein targets—inherent allosteric sites and interaction interfaces—for drug development has been classically viewed as largely distinct approaches. Later, a new generation of drugs was proposed that combined the inherent allosteric regulation of proteins (molecular level) and the protein network data (cellular level), called “allo-network drugs” [26]. Therefore, characterising promising allosteric pockets of the native hVKORC1 and its mutants, as well as the studying of PDI recognition by hVKORC1 mutants, is an urgent task to bring us closer to such developments.

## 2. Results

### 2.1. Modelling and Data Processing

Structural (3D) models of hVKORC1 mutants were built by homology modelling using the experimentally confirmed de novo model of the native enzyme as a template (Figure 1A). These homology models were further refined by conventional MD simulation (all-atom, with explicit water) (Figure 1C). For each hVKORC1 mutant, three independent MD trajectories (replicas 1–3, each of 0.5 µs) were generated under strictly identical conditions to examine their consistency and completeness, and extend conformational sampling. The generated data were analysed for each full-length protein and per domain using either a single or concatenated trajectory. To avoid the protein motion as a rigid body, all data were normalised by least-squares fitting of MD conformations to the initial structure (t = 0 μs). Further, all data generated for the native protein (hVKORC1^WT^) and its mutants were merged and normalised by fitting onto the hVKORC1^WT^ initial conformation (t = 0 µs). These standardised data were used for proteins cross-comparisons. To be able to make such a comparison, we used the same protocol and criteria for data generation of hVKORC1 mutants and their analysis as for native hVKORC1, as described in our previous papers [5,7,19].

### 2.2. General Characterisation of the MD Trajectories

The root-mean-square deviations (RMSDs) and root-mean-square fluctuations (RMSFs) calculated on MD conformations of the individual trajectories of each mutant display comparable profiles, demonstrating the good reproducibility of the generated data (Appendix A). The RMSD values and their variations across all trajectories of hVKORC1^A41S^ and hVKORC1^S52W^ are lower and range from 2 to 6.5 Å. Nevertheless, some trajectories of hVKORC1^W59R^ and hVKORC1^H68Y^ exhibit higher RMSD variations with significantly increasing values (up to 8–10 Å), while other replicas vary slightly, similar to hVKORC1^WT^. The profiles of the RMSF curves are also comparable between the three MD replicas of each mutant, differing only in the values of highly fluctuating N- and C-termini, L-loop, and linkers between the TMD helices, and between the L-loop and TM1. These RMSF profiles are typical of hVKORC1, displaying a saddle-like shape for the L-loop residues, where two maxima are separated by a hollow formed by residues in proximity to S^50^C^51^S^52^. The RMSF maxima differed significantly between trajectories of the same mutant and between mutants.

To compare hVKORC1 mutants with the native protein (hVKORC1^WT^), the RMSF values were recalculated for the concatenated trajectories of each protein and standardised to the same conformation (t = 0 µs, hVKORC1^WT^). Comparing the normalised RMSF curves, we noted that (i) the L-loop saddle relative to the humps is deeper in mutants than in hVKORC1^WT^; (ii) TM helical linkers in mutants fluctuate more compared with those of hVKORC1^WT^; and (iii) maximally fluctuating fragments show different values in hVKORC1 mutants (Figure 2A).

As the L-loop is a key moiety in hVKORC1 activation and carries numerous missense mutations impacting hVKORC1 functions, we mainly focused our analysis on this region comparing R33-N77 aas. The RMSD distributions of the L-loop are described by multimodal Gaussian curves (Figure 2B). Each RMSD curve shows a single highest peak corresponding to the most probable RMSD value specific to each protein studied. The lower probabilistic peaks are either adjacent to the highest peak (hVKORC1^WT^, hVKORC1^W59R^, and hVKORC1^H68Y^) or widely separated (hVKORC1^A41S^, hVKORC1^S52W^) along the RMSD axis. We note that the RMSD curves are most similar in hVKORC1^W59R^ and hVKORC1^WT^ by their profile, the range of values (from 1.5 to 5 Å), and the position of the highest peak (at 2.1 and 2.3 Å respectively). The RMSD probability distribution in other mutants shows a more expanded range of value (up to 7–8 Å). The RMSF probability curves in all mutants represent monomodal flattened distributions with very large variance except hVKORC1^W59R^, which shows, for a large majority of its MD conformations, a well-shaped Gaussian peak ranged from 1.5 to 2.5 Å (Figure 2C).

### 2.3. L-Loop Shape and Conformational Features

Despite the large heterogeneity of RMSD and RMSF distributions, the L-loop size, evaluated by the radius of gyration (Rg), revealed that the majority of the L-loop conformations in all mutants adopt a compact globular shape of comparable size (Rg of 10–11 Å) (Figure 2D,E). However, hVKORC1^A41S^ and hVKORC1^W59R^ display two L-loop sizes; one is slightly reduced and the other increased compared with the compact “closed” L-loop of hVKORC1^WT^. Only a few L-loop conformations of hVKORC1 mutants exhibit an enlarged size (up to 13–14 Å).

Suggesting a dependence between L-loop conformation and shape, we characterised the conformational space of the L-loop explored by each hVKORC1 mutant during MD simulations using ensemble-based convergence analysis [27]. First, the L-loop conformations of each mutant were grouped with different RMSD threshold values varying from 2.4 to 7.0 Å with a step of 0.2 Å (Appendix A). With a cut-off of 4.0 Å, up to 98–100% of the L-loop conformations of all mutants were clustered.

By comparing the population of clusters obtained for the L-loop of each hVKORC1 mutant, we observed that most conformations of hVKORC1^A41S^ are included in two main clusters: highly populated C1 (64%) and half-times smaller C2 (26%); the rest of the conformations form a sparsely populated group (9%) (Figure 3A and Appendix A). L-loop conformations of hVKORC1^H68Y^ are mainly grouped in cluster C1 (69%), and the rest are distributed in three clusters with a comparable population: C2 (12%), C3 (9%), and C4 (8%). L-loop conformations of hVKORC1^S52W^ form two almost equally populated clusters, C1 (37%) and C2 (32%), while the other clusters are less populated: C3 (17%), C4 (9%), and C5 (5%). All hVKORC1^W59R^ L-loop conformations are grouped into two clusters, C1 (62%) and C2(36%).

For each mutant, the representative conformation of each cluster is divergent in folding (2D) and tertiary (3D) structure. Likewise, representative conformations of the most populated clusters of different mutants are also structurally distinct. Nevertheless, the most populated cluster, C1, in all mutants is composed of a compact globular-shaped L-loop having the “closed” conformation, as observed in hVKORC1^WT^ [5,7]. The sparsely populated clusters contain extended L-loop conformations exhibiting an “open” shape, except for hVKORC1^W59R^. In this mutant, both clusters are composed of the “closed” L-loop conformations, differing slightly in the degree of globularity. Similarly to hVKORC1^WT^, numerous intermediate conformations between the “closed” and “open” shapes of the L-loop are observed in the mutants. The predominant globular form of the L-loop is stabilised by an extended network of non-covalent interactions—H-bonds linking residues within the “cap” or “hinge” and van der Waals contacts holding the “cap” and “hinge” in close proximity (Figure 3C). The number of H-bonds in most L-loop conformations typically ranges from 5 and 12 contacts for all mutants, with the number being higher in hVKORC1^WT^, hVKORC1^H68Y^, and hVKORC1^S52W^ and lower in hVKORC1^A41S^ and hVKORC1^W59R^. Van der Waals forces involve two (in hVKORC1^H68Y^ and hVKORC1^W59R^) or four (in hVKORC1^A41S^ and hVKORC1^S52W^) pairs of interacting residues.

To understand the factors contributing to the fixedness of the L-loop, we analysed the occurrence of H-bonds, stabilising the L-loop of each protein as a function of time. We observed that some H-bonds are observed in all mutants and maintained over extensive simulation time. For example, the D38⋯R53 salt-bridge interaction, observed in hVKORC1^WT^, is conserved in the “closed” conformations of the mutants and is stronger in hVKORC1^H68Y^, hVKORC1^S52W^, and hVKORC1^W59R^ (Figure 3B). The loss of such interactions favours the “opening” of L-loop conformations, and such conformations form sparsely populated clusters (e.g., clusters C3 and C4 in hVKORC1^S52W^) (Figure 3B and Appendix A). Such elongated L-loop conformations are generally observed in the simulation time ranges where the RMSD varies significantly and may correspond to transient states of the L-loop.

Backbone–backbone H-bonds L42⋯A48 and T47⋯I49 are conserved in all mutants and stable throughout all MD trajectories. The stabilisation of the “cap” region, along with the S−S covalent bridge, is maintained by salt–bridge interactions and backbone–backbone H-bonds L42⋯A48 and T47⋯I49. The observed H-bonds, shown in Figure 3C, explain the cohesion of each L-loop module, “cap” and “hinge”, in each mutant, as in hVKORC1^WT^ [5,7].

The “closed” shape of the L-loop is maintained by non-covalent interactions between the hydrophobic residues of the two L-loop modules. The observed patterns of intramolecular contacts caused by van der Waals forces in the hVKORC1 L-loop evidenced their key role in maintaining two L-loop fragments in close proximity. These results show that intramolecular interactions (van der Waals and electrostatic forces) are dominant factors in stabilising the compact “globule-shaped” L-loop. Such intermolecular forces, at the origin of the globularity of the L-loop, prevent its expansion (“opening”) caused by the solvent.

### 2.4. L-Loop Folding and Plasticity

Similarly to the native enzyme in the oxidised state, the helical folding in mutants involves nearly every tier of L-loop residues with mean values of 30, 34, 27, and 27% in hVKORC1^A41S^, hVKORC1^H68Y^, hVKORC1^S52W^, and hVKORC1^W59R^, respectively. Such a fold is exhibited by three small (3–4 residues) transient helices H1-L, H2-L, and H3-L, partially converting between αH- and 3_10_-helices (Appendix A and Figure 4B). In most cases, the helical content in mutants is close to that in hVKORC1^WT^; nevertheless, its probability is different for each protein studied (Figure 4A). The helical content probability curves of hVKORC1^A41S^, hVKORC1^S52W^, and hVKORC1^WT^ show the best superimposition. Their most probable folding is identical (28%) and the probability of L-loop conformations having less or more folded structure is different only slightly. The helical content of most L-loop conformations in hVKORC1^W59R^ varies from lower (18%) to higher (35%), with a most probable value of 28%. In hVKORC1^H68Y^, two main L-loop conformations with helical contents of 18 and 23–27% are observed.

Clustering of L-loop conformations based on the secondary structures showed that H1-L is the most conserved among all mutants by size and appears mainly as an α-helix, except in rare conformations in which the α-helix does transit into a small 3_10_-helix (Figure 4B). The H2-L helix is fully transient (α-helix ↔ 3_10_-helix), and its 3_10_-helix fraction is predominant. The reversibly transient H2-L, seen as shortened or elongated in length, is a single or sequential double helix, with a 3_10_-helix fraction varying in the mutants: hVKORC1^A41S^ > hVKORC1^S52W^ > hVKORC1^W59R^ ≈ hVKORC1^H68Y^. Likewise, H3-L is completely transient, exhibiting distinct fractions of two types of helices in different mutants. In hVKORC1^S52W^, H3-L also transits into a coil (α-helix ↔ 3_10_-helix ↔ coil). The combination of helical fractions is a fundamental criterion for grouping the L-loop conformations according to their folding.

The positions of the L-loop helices in the mutants are overall conserved in sequence, as was observed in the native enzyme [5]. To illustrate the relative orientation of the L-loop helices in 3D space, their spatial dynamical drift was analysed. The axis of each helix was defined for MD conformations from concatenated trajectories (sampling every 100 ps), superimposed, and projected onto a randomly chosen L-loop conformation (Figure 4C). The extended distributions of superimposed axes (elongated by 50% to better represent their position and direction) for all helices form extended distributions, showing both the most probable and the rarest positions.

The broad distribution of helices in the mutants differs from the dense and compact reap-like distributions of hVKORC1^WT^ [5]. The axis of each helix varies considerably in its spatial orientation within its distribution for a given mutant, as well as for the same helix between different mutants, reflecting the mutant-specific drift of each helix.

In mutants, the L-loop helices’ drift is ensured by their highly flexible linkers (see RMSF), which, together with the linker connecting the L-loop to TM1 from TMD, provide higher helix mobility compared with hVKORC1^WT^. H1-L, mostly folded into a regular α-helix, contains C^43^ that is covalently linked to C^51^, located on the coil connecting the H1-L and H2-L helices. Such an S−S covalent bond significantly restricts the conformational mobility of the “cap” region. A large, coiled linker connecting the H2-L and H3-L helices favours the large movement of these helices.

This observation, at first glance, seems inconsistent with the preferred “closed” L-loop conformation. We note that in several mutants, the TMD helix fragments, located near the L-loop, exhibit mutant-induced partial unfolding (Appendix A). In particular, in all mutants, the upper segments of TM2 and TM3 adjacent to the L-loop are partially disordered, exhibiting reversible transitions (α-helix ↔ 3_10_-helix or/and α-helix ↔ 3_10_-helix ↔ coil).

This partial unfolding of the TMD helices increases their flexibility, which affects the plasticity of the L-loop, as evidenced by increased drift of the L-loop helices. We hypothesised that the drift of the L-loop helices and the movement of the upper segments of TM2 and TM3 may be highly correlated.

### 2.5. Inherent Dynamics of hVKORC1 Mutants

To characterise the plasticity of the studied proteins, the inherent dynamics of hVKORC1 mutants were first analysed by estimating the collective motions using Principal Component Analysis (PCA) performed in Cartesian coordinates and a singular value decomposition approach [28]. Calculations were performed on Cα atoms only. The first two PCA components (modes) denote high mobility of the terminal residues (N- and C-term) (Appendix A), which was also evidenced by their highest RMSF values. Since the N- and C-terms (M1-G9 and K152-H163 aas) are not the focus of our study, they were excluded from further computation of the explained variance by different principal components (PC).

Ten modes, characterised by the cumulative and individual explained variances, describe about 90–95% of the variance of hVKORC1 containing TMD and the L-loop, or only the L-loop (Figure 5A). Projecting MD conformations into phase space along the first three principal eigenvectors (PC1, PC2, and PC3) revealed that (i) the sampled conformational space is largely extended in all mutants versus a more compact area in hVKORC1^WT^; (ii) the hVKORC1 conformations provide poor overlap between hVKORC1^WT^ and its mutant subspaces, especially for the L-loop (Figure 5B).

The first two principal components, PC1 and PC2, characterise most of the movements in all mutants. PC1 is the direction along which the projections have the largest variance, and PC2 is the direction which maximises the variance among all directions orthogonal to the first. These two PCs adequately characterise the degree of inharmonicity of the inherent molecular dynamics of each studied protein. The atomic displacements in the first two PCA modes were projected onto the respective average structures to visualise the direction and amplitude of the principal movements (lowest frequency, most collective motion) (Figure 5C,D).

In the studied proteins, all structural elements of the L-loop, including helical and coiled linkers, are involved in large-amplitude collective movements. However, the motion of the L-loop in mutants differs from the scissor-like movements observed in hVKORC1^WT^, and also alters between mutants. The two L-loop fragments—the “cap”, stabilised by disulfide bridge and H-bond interactions, and the significantly relaxed “hinge”—exhibit a wide range of motions, oriented either in the opposite or orthogonal direction relative to each other or alongside it. The amplitude and direction of these movements also differ between distinct hVKORC1 mutants. The scissor-like movement of the H1-L helix (“cap”) and the linker connecting H2-L and H3-L (“hinge”) found in hVKORC1^WT^ is also observed in hVKORC1^S52W^, but the movement of “hinge” is decomposed into two components, having orthogonal directions and increased amplitudes. The “hinge” motion in other mutants is also broken down into two or more components, characterised by alternating local displacements that differ in direction and amplitude. This decomposition of the “hinge” motion is closely connected to the large collective motion of the TM helices, an effect not observed in hVKORC1^WT^. In hVKORC1^WT^, the structurally stable and almost static TMD helices participate only in collective drift within the membrane [19], whereas in mutants, the TMD helices exhibit ample motion of their regions adjacent to the L-loop.

To elucidate the interplay between the L-loop and TMD movements, cross-correlation matrices were calculated for the Cα atom pairs of each hVKORC1 mutant. The pairwise Cα–Cα distance patterns demonstrate the coupled motions within each structural domain, L-loop and TMD, and between these domains (Figure 5E). We noted that (i) the fine-grained muted patterns of hVKORC1^H68Y^, hVKORC1^S52W^, and hVKORC1^WT^ are composed of well-defined blocks, generally corresponding to the structural elements (helices or coils) of protein, and exhibiting positive or negative correlations; (ii) the hVKORC1^A41S^ and hVKORC1^W59R^ cross-correlation patterns are noticeably brighter and composed of enlarged grain blocks that frequently overlap between different structural units (helices and coils) and/or domains (TMD and L-loop). Focusing on the L-loop, its inherent cross-correlation pattern is also different between the studied proteins. In particular, we observed (i) a distinct red block demarcating L-loop residues, subdivided into two sub-blocks, showing a positive correlation of its structural units—“cap” and “hinge”—only in hVKORC1^H68Y^ and hVKORC1^WT^; (iii) the red L-loop block is significantly reduced in hVKORC1^A41S^ and hVKORC1^S52W^ and increased in hVKORC1^W59R^.

The coherence of the motions of the L-loop “hinge” and TM2 helix promotes a reduction of the L-loop fraction in the pattern, showing independent dynamics in hVKORC1^A41S^ and hVKORC1^S52W^, while the TM1 contribution to motion consistent with L-loop movement leads to an increase in the dynamical coupling of the L-loop and TMD in hVKORC1^W59R^. Such an effect was observed early in the metastable reduced states of hVKORC1^WT^ occurring during the enzymatic cycle [19].

Focusing on the relationships between the point mutations and other residues of hVKORC1 and comparing them with the native protein, we note that the short- and long-range effects significantly alter the correlation pattern (Figure 5F). Mutation impacts the coupling of proteins residues which manifest through both local (short-range) and widespread (long-range) scaling effects evidenced at the sequence level and 3D space (tertiary structure).

For instance, A41S, positioned on the linker connecting TM1 and the L-loop, positively correlates with the H1-L helix (L-loop). This correlation shows an increase in local coupling within the “cap” of the L-loop, exhibiting higher collective motion in the mutant compared to the native protein. Additionally, the A41S mutation alternatively correlates with each membrane helix. Thus, its correlation is negative with the upper half of TM1, the whole of TM2 and the middle part of TM3, while with TM4, only positive correlations are observed. The S52W mutation diminishes and partially alters the correlations between all residues of the mutant with respect to hVKORC1^WT^. The H68Y mutation did not significantly alter local coupling compared to the native protein; in both proteins, all residues of the L-loop are positively correlated. In contrast, this mutation influences dynamical relationships with TM2 (negative correlations) and TM3 (positive correlations). In hVKORC1^W59R^, the W59R correlates positively with half of the protein, involving the L-loop, TM1, TM2, and the loops connecting TM1 and TM2, and negatively with TM3 and TM4. Thus, W59R is the most troublemaking mutation of hVKORC1, disturbing the dynamics of half of the protein, whereas the other mutations only affect a quarter. Such a comparison allows us to confirm again that in all proteins studied, the missense point mutations of key residues affect the DYNASOME of the protein and this influence is sequence-dependent.

### 2.6. The Free Energy Landscape as a Quantitative Measure of the Mutation-Induced Effects in hVKORC1

To compare the conformational space of hVKORC1 mutants, we adopted a robust strategy for the in-depth analysis: generating a free energy landscape along specifically chosen coordinates called “reaction coordinates” or “collective variables” that describe the conformation of a protein [29,30,31]. The intrinsic protein energy landscape can be quantified using the density of states or the statistical energy distribution of conformations, which can be quantified by transforming the canonical ensemble representation to a microcanonical one.

Such interpretation leads to quantitatively significant results that allow comparison of different forms and states of proteins or their mutants. The relative Gibbs free energy ΔG is a measure of the probability of finding the system in a given state. This representation of the protein sampling by using reaction coordinates can be a quintessential model system for assessing barrier crossing events in proteins [32,33]. In general, this process can be estimated from an incomplete sampling of the protein states, if it is an unbiased sampling.

In our case, using rich numerical data obtained by merging all generated trajectories for hVKORC1^WT^ and its four mutants, we attempted to compare the MD conformations of different hVKORC1 proteins that differ by only one residue. Since all proteins were simulated under identical conditions, these data, after normalisation to the same conformation, can also be interpreted in terms of the relative free energy.

To generate the free energy cumulative landscape (FECL) of hVKORC1 and its mutants, we used the first two PCA principal components (PC1 and PC2, describing about 90–95% of variance) as reaction coordinates. The FECL constructed on PC1 versus PC2 shows a rugged landscape spectrum, revealing the high conformational diversity of the L-loop, reflected by the multiple well-defined minima resulting from the multimodal distribution of both reaction coordinates metrics (Figure 6A,B).

The deepest W1, together with the adjacent low minimum W4, forms an almost common conformational space consisting of two close subspaces, separated from the others by very high energy barriers. This FECL profile is derived, on one hand, from the bimodal distribution of the PC1 component, showing a well-defined sharp probability maximum, complemented by a flat extended peak, and, on the other hand, from a series of maxima in the multimodal distribution of PC2 with a single distinct higher peak. As a result, the deepest W1 on the hVKORC1 FECL is completed by a series of minima represented by lower adjacent wells W2–W4 and a distant low minimum W5, separated from the other wells in the FECL by very high energy barriers.

The Gibbs free energy landscape, exhibiting multiple minima, was searched for its protein content. Each well on the FECL includes different content, consisting of MD conformations of two or three proteins (W1, W2, and W4) or a single protein (W4 and W5) (Figure 6C,D). We explored the content of each well regarding the protein composition. The wells W1, W2, and W4 are multi-protein sub-ensembles that compile from the MD conformations of different proteins. In particular, the hVKORC1^WT^ MD conformations are the major component (56%) of W1, coexisting with the hVKORC1^H68Y^ mutant (23%); the other mutants are also present in W1, but in a minor quantity (4–9%). Similarly, W2 is principally composed of MD conformations of two proteins, hVKORC1^A41S^ (60%) and hVKORC1^W59R^ (35%); the hVKORC1^WT^ MD conformations (5%) complete the W2 sub-ensembles. The three-component W4 contains hVKORC1^H68Y^ (48%), hVKORC1^WT^ (37%), and hVKORC1^S52W^ (15%) MD conformations. In contrast, W3 and W5 contain mono-protein sub-ensembles composed of only hVKORC1^W59R^ (95%) and hVKORC1^S52W^ (100%) MD conformations, respectively.

From this analysis, some general conclusions may be postulated: (i) the minimum-compactness energy landscape spectrum corresponds to a conformational ensemble composed of the native protein hVKORC1^WT^ and mutants having the major native-like shape and size of L-loop (‘closed’ conformation with Rg of 10.7 Å); (ii) W2, proximal to W1, is composed mainly of two mutants, hVKORC1^A41S^ and hVKORC1^W59R^, possessing a more compact globular L-loop (Rg of 10.2 Å) with a highly conserved H1-L helix; (iii) W1 and W2 contain only “closed” L-loop conformations differing mainly at the level of their compactness; (iv) W3 and W5 minima are composed of less “closed” and “open” L-loop conformations of hVKORC1^W59R^ and hVKORC1^S52W^, respectively; (v) W4 compiles all less “closed” L-loop conformations of all proteins, except hVKORC1^A41S^ and hVKORC1^W59R^ in which almost L-loop conformations are “closed”.

Our results indicate that the minima of the free energy landscape reflect internal conformational and structural features of the L-loop. L-loop conformations of the same shape and size form overlapping subspaces even from different proteins, both native and mutated. Distinguishing and isolating the conformational ensemble of each disordered protein differing from others by a point mutation is a not trivial task, even if one uses PCA-derived metrics describing the largest variances of the studied proteins.

### 2.7. Pocket Search in the Native hVKORC1 and Its Mutants

Our previous studies of hVKORC1 focused either on developing topological and 3D models of its functionally related states (inactive, activated, and active) formed during the catalytic cycle [19] and studying the mutation-induced effects on the structural and dynamic properties of hVKORC1 in the inactive (oxidised) state by comparison with the native protein (present work). The characterisation of conformational ensembles using structural, dynamics, and free-energy descriptors was compiled as the DYNASOME defining hVKORC1 as the modular intrinsically disordered protein [5,7]. In these studies, we have not addressed one of the most interesting aspects of hVKORC1: its binding pockets, playing a crucial role in the physiological functioning of this enzyme. Some pockets are responsible for the recognition and binding of its redox protein, and others for the binding of vitamin K and AVKs. Although hVKORC1 state-selective recognition of vitamin K in its different forms (epoxide, semiquinone, and quinone) and their binding in the active site pocket have been studied [19], as well as recognition of hVKORC1 by PDI [5], the pockets of the inactive (fully oxidised) state are still not sufficiently characterised.

The structural and dynamic variations in intrinsically disordered proteins lead to a large set of different conformations (conformational space), which often result in the formation of “cryptic” or “hidden” pockets that require a conformational change to be apparent [34]. To explore the putative cryptic pockets in hVKORC1, we first searched the MD simulation data by testing whether or not the cryptic pockets are observed in the inactive state of each protein studied.

A pocket search (using MDpocket [35]) of the full-length hVKORC1^WT^ and the L-loop alone revealed the well-known active site pocket and allosteric pocket on the L-loop (Figure 7A). The found pockets were numbered according to their volume (V)—the largest P1, extending from the L-loop S-S bridge across the entire upper half of the TMD, and a smaller P2 located between the L-loop “cap” and “hinge”.

The pocket P1 in the inactive state of hVKORC1^WT^ is closed to vitamin K entry (vitamin K gateway) by an L-loop S-S bridge. Its elongated tunnel-like shape with a volume of 600–1050 Å^3^ likely shows the circuit positions that vitamin K visits, starting from its entry into activated hVKORC1, its positions in the active site where the vitamin K is transformed by the active hVKORC1 from the epoxy form to semi-quinone and then into quinone, which leaves the enzyme.

Because we believe that the L-loop is a domain of hVKORC1 suitable, in isolated form, for in silico and in vitro studies, the L-loop pockets were characterised only for this cleaved module alone. The pocket P2 shows an almost spherical shape of a volume varied from 50 to 230 Å^3^ (Figure 7B,C). Tiny P3 (0 to 100 Å^3^) corresponds to the upper part of the P1 pocket, starting from the L-loop.

The pocket P2 is formed by the charged (R35, D36, R53, R61 and E67), polar (S57 and T47), and hydrophobic (C43, I49, and L65) residues (Figure 7D). Similarly, the P3 surface is also formed by highly heterogeneous residues (R37, S52, R53, R61 and I75). In the analysis of the hydrophobicity score, the essential pocket descriptor [36], in a function of the pocket volume, we note that the maximal size of P2 and P3 corresponds to the hydrophobicity score of 10-15. Nevertheless, the detailed analysis of conformations having the maximal volume of pockets P2 (460–506 Å^3^) and P3 (184–220 Å^3^) showed the pocket score ranging from 1 to 27 and from 2.6 to 15.3), respectively.

While arginine (R) residues—R35, R53, and R61—contribute to the formation of either one L-loop pocket or both, we analysed the volume of two pockets as a function of arginine quantity at the pockets’ surfaces (Figure 7F). The P2 surface contains from one to three arginine residues (R35, R53, and R61), and the pocket volume apparently depends on the orientation of their side chains. Indeed, in smaller P2, the side chain of R35 is oriented outside of the pocket cavity, while the side chains of R53 and R61 are positioned inside the cavity and contribute to stabilising the “open” conformation of the L-loop. In contrast, if the side chains of all arginine residues are outside of the P2 pocket cavity, the volume of P2 is considerably increased, as observed in the “closed” L-loop conformation. The surface of the small P3 pocket comprises one or two arginine residues, R35 and R61. The smallest volume of P3 is observed in the “closed” L-loop conformations with R35 and R61 side chains oriented outside and inside the pocket, respectively. The “open” L-loop conformations with a similar outside orientation of R35 and R61 exhibit a larger P3 cavity.

The orientation of the arginine side chain is not a single factor impacting pocket volume. A similar effect may be occurring from the inside-outside orientation of the other residues’ side chains, contributing to the pockets’ surface formation (Figure 7D).

The analysis of pockets in hVKORC1 mutants clearly demonstrates the mutation-induced effects on their appearance, localization, and size (Figure 8).

First, the large tunnel-like P1 of hVKORC1^WT^ is transformed in each mutant into a spherical pocket located at the S-S bond formed by the catalytic cysteine residues C^132^ and C^135^. The pocket volume is significantly reduced (by a factor of 2–3) in all mutants compared with hVKORC1^WT^. Moreover, in addition to this pocket, in hVKORC1^A41S^ and hVKORC1^W59R^, two tiny pockets, P2 and P3, are found. These pockets do not correspond to the P2 and P3 pockets in the native protein, but rather are products of the disrupted tunnel-like pocket P1 of hVKORC1^WT^, as indicated by their positions.

As for the general identity (status) of the detected pockets, both in native hVKORC1 and in its mutants, they are all transient and vary from 0 to a maximum value, which depends on the pocket and the protein. Therefore, the intrinsically disordered hVKORC1 in the native and mutated forms possesses cryptic pockets which may expand the scope of drug discovery of these targets.

### 2.8. Probing of the PDI Recognition by hVKORC1 Mutants

To complete the comparative characterisation of hVKORC1 mutants with the native protein, we explored the protein disulphide isomerase (PDI) binding to each mutant. We probed only the modelling of the non-covalent molecular complex preceding the thiol-disulfide exchange reaction, a state described by the precursor complex. To dock PDI and each mutant, we used the protein–protein automatic docking with High Ambiguity Driven protein–protein DOCKing (HADDOCK) [37,38]. Unlike other protein–protein docking approaches, based on the combination of energetics and shape complementarity, HADDOCK uses biophysical interactions data—in our case, a short distance between sulphur atoms from the cysteine residues of two interacting protein, PDI and hVKORC1, to drive the docking process. We successfully used this approach to build a de novo model of the first precursor of PDI/hVKORC1 molecular complex formed by native hVKORC1 and its redox protein PDI. We are now asking: do missense mutations influence PDI recognition by hVKORC1? To achieve this, we carefully performed the docking using protocol parameters and procedures previously used for the native protein.

The docking solutions and their clustering show that two mutants, hVKORC1^A41S^ and hVKORC1^W59R^, docked onto PDI, systematically reproduced the PDI/hVKORC1^WT^ precursor complex (Figure 9 and Appendix A).

The docking solutions for two other mutants represent models of complexes PDI/hVKORC1 in which the redox protein is inserted into the membrane. In addition, in the PDI/hVKORC1^H68Y^ model, the PDI protein shows significant infolding. Such models are incompatible with the environmental conditions and the protein functions.

## 3. Discussion

Continually emerging variability of pathogenic proteins frequently leads either to alternation of the protein activity or resistance to currently used drugs, which pose a huge challenge for the treatment of many diseases. It is critical to comprehensively characterise the mutation-induced effects and decipher their mechanisms. Such a description of mutated proteins is frequently performed by structure-based computational methods, producing spatiotemporally linked information on the mutation-induced structural effects at the atomistic level. Apart from the knowledge that sheds light on a deep comprehension of dysfunctionality or aberrant functionality of the missense variants (fundamental learning in biology), the in silico data produced are largely used for the development of novel drugs to combat resistance effects or prevent possible evasion of resistance tactics by pathogens (pharmacological or clinical application of learning).

In our work, we focused on the human Vitamin K Epoxide Reductase Complex, an important therapeutic target. The continued emergence of hVKORC1 strains with reduced susceptibility to available anticoagulants (VKAs) is a major public health concern that threatens ongoing efforts against blood clotting disorders in humans [8,12] or expansion-resistant rat populations [39,40]. In some cases, genetic polymorphism induced either impaired enzymatic ability or the complete absence of catalytic property, leading to the failure of metabolic and biochemical reactions in tissues [41]. For a small VKORC1 protein (163 aas), a total number of 193 variants was reported in the LOVD^3^ database [42]. Since factors contributing to this variability include age, vitamin K intake, concomitant medications, acute and chronic disease status, ethnicity, and genetics [43,44,45], the effects observed in patients treated with vitamin K antagonists are highly heterogeneous [46].

Given these challenges associated with ambiguous interpretation of empirical data, numerical characterisation of hVKORC1 should be (i) separated as much as possible from uncertain/contradictory literature data, and (ii) based on methods that produce pragmatic and rational results, a strategy that we already used for a study of this protein [7,19]. Moreover, with such a great number of hVKORC1 missense variants and the intrinsic disorders of this protein, using in silico methods is the most appropriate approach to obtain crucial, high-quality information. The reliability of using in silico methods for hVKORC1 characterisation has already been successfully demonstrated by the generation of de novo 3D models of a protein with a four-helix transmembrane domain and an intrinsically disordered L-loop [19], which were confirmed by crystallographic structures published four years later [20].

The multifaceted description of a protein by compiling data describing its chemical composition (sequence, 1D), local folding (secondary structures, 2D), global architecture (tertiary structure, 3D), and dynamics (conformational ensemble) completes its DYNASOME [21]. Such a description provides a solid basis for studying the relationships between the DYNASOME descriptors and the protein activities (e.g., signalling and enzymatic reactions). During these physiological processes, a protein forms metastable protein–protein molecular complexes that constitute its INTERACTOME [47]. Protein–protein binding occurs through the surface allosteric binding pockets, which are topographically distinct from the orthosteric site that binds the natural ligands, forming the POCKETOME of a protein [22,48]. In sum, this multifaceted “omics” (multi-omics) characterisation of a protein is a crucial cornerstone for the understanding of biological processes in which a given protein participates.

In our study, we focused mainly on the luminal loop (L-loop), the principal mediator of hVKORC1 activation, and a region often carrying numerous missense mutations. Four L-loop hVKORC1 mutants, possessing missense mutations A41S, W59R, S52W, and H68Y, were studied by numerical approaches (in silico) and compared with the native protein. All proteins were considered in the fully oxidised (inactive) state. Although these four hVKORC1 mutants were associated in [8] with the resistance phenotype observed in patients (hVKORC1^A41S^ and hVKORC1^H68Y^) or leading to a loss of hVKORC1 activity (hVKORC1^S52W^ and hVKORC1^W59R^), their selection for the multi-omics characterisation was based on a purely logical concept that primarily focuses on differences in physico-chemical properties between native and mutated residues, as well as the locations of missense mutations, which are positioned either in the “cap” (A41S and S52W) or in the “hinge” (W59R and H68Y). Since the S-S disulphide bridge formed by C^43^ and C^51^ in the “cap” is involved in the direct thiol-disulfide interconversion between hVKORC1 and its redox protein PDI, and the fully relaxed “hinge” is involved in the recognition of the PDI [5], we hypothesised that the chosen missense mutations may provoke effects detectable at all levels of multi-omics description.

All studied proteins, the native hVKORC1 and its mutants, show similar topology but different structures and dynamics (DYNASOME). The L-loop missense mutations affect the folding and dynamical properties of the L-loop itself, as well as the TMD. Similarly to hVKORC1^WT^, the compact globular-like “closed” L-loop conformation is the most common of all mutants, while the extended “open” shape is observed either rarely (hVKORC1^A41S^ and hVKORC1^W59R^) or never (hVKORC1^S52W^ and hVKORC1^H68Y^). Despite the similar shape, the L-loop conformations in the proteins studied exhibit significantly different folding (2D structure), a propriety typical of intrinsically disordered proteins and apparently mutant-dependent, since this compact globular conformation of the L-loop is maintained mainly by van der Waals contacts, specific for every mutant. Interestingly, transient structural elements of the L-loop (α-helix, 3_10_-helix, coil) and their combination, and their relative orientation in 3D space, are also mutant-specific.

The DYNASOME characterisation of hVKORC1 mutants showed that L-loop missense mutation effects indicate a strong interdependence of the L-loop and the TMD not found in the native protein. The great dissimilarity between the DYNASOME of the native protein and its mutants is evidenced by the mutant-induced partial unfolding of TM2 and TM3 TMD helices in the fractions adjacent to the L-loop. This structural disorder increases the flexibility of these helices, affects the plasticity of the L-loop, and alternates correlations of the inherent dynamics in hVKORC1 mutants with respect to the native protein. According to our analysis, the W59R is the most troublemaking mutation of hVKORC1, disturbing the dynamics of half of the protein, whereas the other mutations only affect a quarter of the protein.

As we reported previously, the structural and dynamics metrics of native hVKORC1, simulated either as the protein inserted into a membrane [5,19] or placed in an aqueous solution [7], are similar. These results indicate that hVKORC1 is composed of the stable transmembrane domain and the disordered L-loop, located in the endoplasmic reticulum lumen. Furthermore, the de novo predicted model of hVKORC1 matches well with crystallographic structures (no membrane) [20]. Based on these results, the mutated proteins were simulated in an aqueous solution. However, since we observed that the missense mutations affect not only the L-loop but also part of the transmembrane helices, it will be interesting in perspective to analyse these mutants as inserted into the membrane to distinguish between the mutation-induced effects and those promoted by the membrane.

To reveal the underlying relationships between the DYNASOME–POCKETOME–INTERACTOME of the five hVKORC1 proteins, we characterised (i) their sequence–structure–dynamics properties, which evidenced their mutant-specific dependence, (ii) compared the topography of these intrinsically disordered proteins through the free energy landscape, (iv) localised their cryptic pockets, and (v) explored them as PDI targets.

Observing such a large number of mutation-induced effects through the DYNASOME–POCKETOME–INTERACTOME description, and their clear sequence-dependence, we attempted to compare these proteins and distinguish them by generating a free energy landscape. Since the motion of these proteins is a central factor determining their DYNASOME properties, a cumulative free energy landscape was constructed on two principal components describing the vast majority of hVKORC1 movements. Because all proteins studied are intrinsically disordered, we did not expect to obtain completely separated global minima for each protein. However, we observed two local minima (W3 and W5) consisting of the conformations of only one protein, hVKORC1^W59R^ and hVKORC1^S52W^, respectively. The deepest wells, W1, W2, and W4, include different content, consisting of MD conformations of two or three proteins.

Our results indicate that the minima of the hVKORC1 free energy landscape reflect rather the inherent conformational and structural features of the local L-loop. The most differing mutant-specific L-loop conformations are compiled in distant wells; however, L-loop conformations of the same shape and size form overlapping subspaces even from different proteins, both native and mutants. Distinguishing and isolating the conformational ensemble of each disordered protein, which differs from others by point mutation, is a not trivial task, even if one uses PCA-derived metrics describing the largest variances of the studied proteins.

Another consequence of the mutation-induced effects on protein DYNASOME manifests in the global changes (geometric, topological, and probabilistic) of the inherent pockets. Structural changes drive the formation and evolution of cryptic sites. The highly varied L-loop conformations in the native protein promote the formation of pockets varying in their volume, and belonging to cryptic pockets [34], detected during MD simulations [49]. In some hVKORC1 mutants, these pockets disappear completely, while in others, they appear as new pockets. Although the globular “closed” conformation of the L-loop is the most likely among all hVKORC1 studied, we observed an alteration of the recognition properties of the L-loop with respect to PDI, the hVKORC1 redox protein. According to our preliminary studies (docking trials), two mutants, hVKORC1^A41S^ and hVKORC1^W59R^, recognise PDI similarly to the native protein, while two others, apparently, are not capable of binding this redox protein. Since the folding and binding are tightly coupled, the changes in folding of the L-loop in hVKORC1^S52W^ and hVKORC1^H68Y^ may be the crucial factors influencing the PDI recognition and binding.

Since we studied the inactive form of the protein, we cannot draw any conclusions regarding the sensitivity or resistance of the mutants to AVKs (warfarin or its analogue), since this property is manifested exclusively by the activated protein. Likewise, it is not possible to estimate the correctness or predictability of the results obtained by docking trials. To our knowledge, the ability or incapacity of hVKORC1 mutants to recognise and bind its redox protein has never been reported in the literature.

Based on our results, we postulated that (i) the intra-protein allosteric regulation is specific to the native hVKORC1 and each of its mutants; (ii) the inherent allosteric regulation and cryptic pockets of each mutant depend on its DYNASOME; (iii) the recognition of the redox protein by hVKORC1 (INTREACTOME) depends on their DYNASOME. This multifaceted description of proteins produces ”omics” data sets, which are the crucial keys to understanding the physiological processes of proteins and the pathologies caused by degradation at any of these “omics” levels. Additionally, such characterisation opens novel perspectives for the development of “allo-network drugs” necessary for the treatment of blood disorders.

This innovative concept of “allo-network drugs” is based on two types of inhibition: intra-protein competitive or allosteric inhibitors and inter-protein modulators interacting at the protein-protein interaction interface [26]. Such an approach is a way to improve treatment by increasing the drugs’ specificity, avoiding or significantly reducing the side effects caused by non-specific molecules, and possibly limiting the rapid evolution of new protein strains.

## 4. Materials and Methods

### 4.1. Homology Models

**Homology models**: 3D homology models of the full-length hVKORC1 (1–163 aas) mutants—hVKORC1^A41S^, hVKORC1^H68Y^, hVKORC1^S52W^, and hVKORC1^W59R^—were generated with Modeller [50]. The de novo model of hVKORC1 in a fully oxidised state [19] was used as a template. The stereochemical quality of 3D models was assessed by Procheck [51], which revealed that more than 96% of nonglycine/nonproline residues have dihedral angles in the most favoured and permitted regions of the Ramachandran plot, as is expected for good models.

### 4.2. Molecular Dynamics Simulation

#### 4.2.1. Set-Up of the Systems

The homology models of each mutant, were prepared with the LEAP module AMBER Tools 20 (http://ambermd.org/AmberTools.php) [52] using the ff14SB all-atom force field parameter set [53] and TIP3P water models: (i) hydrogen (H) atoms were added; (ii) covalent bond orders were assigned; (iii) protonation states of amino-acids were assigned based on their solution for pK values at neutral pH, and the histidine residues were protonated on their ε-nitrogen atoms; (vi) counter-ions (Cl^−^) were added to neutralise the charge of each protein; (v) each protein was placed in an octahedron water box.

#### 4.2.2. Minimisation and Equilibration of the Systems

Each system consisting of a protein surrounded by water was minimised and equilibrated using the Sander module of AmberTools20 (http://ambermd.org/AmberTools.php) using the steepest descent and conjugate gradient algorithms through the 30,000 minimisation steps as follows: (i) 10,000 minimisation steps where water molecules have fixed, (ii) 10,000 minimisation steps where the protein backbone is fixed to allow protein sidechains to relax, and (iii) 10,000 minimisation steps without any constraint on the system. A 100 ps thermalisation step was performed, where the temperature (atomic velocity) is gradually increased from 0 to 310 K using the Berendsen thermostat with imposed periodic boundary conditions and isotropic position scaling [54]. Then, a 100 ps equilibration with constant volume (NVT) and a 100 ps equilibration with constant pressure (1 bar) (NPT) were performed. For these two steps, temperature regulation was performed with Langevin dynamics with friction coefficient γ = 1. Finally, a 100 ps molecular dynamics was completed at 310 K (Langevin dynamics), constant volume, and constant pressure (hybrid Monte-Carlo barostat) [55]. All equilibration steps were carried out with an integration step of 2 fs. Non-bonded interactions were calculated with the Particle Mesh Ewald summation (PME) with a cut-off of 10 Å and bonds involving hydrogen atoms were constrained with the SHAKE algorithm [56].

#### 4.2.3. Production of the Conventional Molecular Dynamics Trajectories

The conventional Molecular Dynamics (MD) trajectories of each homology model of mutated hVKORC1 were generated using the AMBER ff14SB force field with the PMEMD module of AMBER 16 and AMBER 18 (GPU-accelerated versions) [52] proceeding on a local hybrid server (Ubuntu, LTS 14.04, 252 GB RAM, 2x CPU Intel Xeon E5-2680 and Nvidia GTX 780ti) and the supercomputer JEAN ZAY at IDRIS.

A time step of 2 fs was used to integrate the equations of motion based on the Leap-Frog algorithm [57]. The Particle Mesh Ewald (PME) method, with a cut-off of 22 Å, was used to treat long-range electrostatic interactions at every time step. The van der Waals interactions were modelled using a 6–12 Lennard–Jones potential. The initial velocities were reassigned according to the Maxwell–Boltzmann distribution. For the molecular complexes, to prevent the separation of the PDI protein from hVKORC1 and to bring them together, the distances between a restrained harmonic distance were introduced to the S⋯S atom pair (the sulphur atoms from C37 of PDI and C43 of hVKORC1), which was varied in a stepwise manner as in [5]. During each step (100 ns), the constraints were maintained, and then removed, to fully relax the systems. For each system, three 0.5 µs trajectories were carried out with different starting velocities. The coordinates were recorded every 10 ps.

### 4.3. Protein-Protein Docking

Protein–protein docking was performed with the HADDOCK2.4 web server (https://wenmr.science.uu.nl/haddock2.4/). HADDOCK (High Ambiguity Driven protein-protein DOCKing) [37] is a protein−protein docking approach based on available biochemical or biophysical information to drive the docking process. The docking protocol consists of several steps with user-defined input parameters. First, the topology and coordinate files are generated for each molecule separately, and merged to generate the starting models of the complex. Second, 10,000 structures were randomly sampled and subjected to rigid body energy minimisation (it0). Third, the best 200 structures were selected, and a semi-flexible simulated annealing in torsion angle space was performed on them (it1). Finally, the obtained structures after the previous step were refined in Cartesian space with explicit solvent (TIP3P)—a short molecular dynamics stage. After, the water-refined structures were clustered using a 7.5 Å RMSD cut-off and sorted according to the HADDOCK score. The maximum number of clusters was set to 10 and the minimal cluster size was set to 4. All other input parameters were kept default. To guide the docking, a set of ambiguous interactions restraints (AIRs), a pair of cysteine residues, C43 of hVKORC1 and C37 of PDI, was provided.

### 4.4. Data Analysis

Standard analysis. Unless stated otherwise, the data analysis was performed using the CPPTRAJ 4.25.6 program [58] of AmberTools20 (http://ambermd.org/AmberTools.php) for MD conformations taken every 10 ps of simulation after least-square fitting [59,60] on the initial conformation (t = 0 ns) of a protein or region of interest, thus removing rigid-body motion from the calculations.
RMSD and RMSF values were calculated for the Cα-atoms using the initial model (at *t* = 0 ns) as a reference.Secondary structural propensities for all residues were calculated using the Define Secondary Structure of Proteins (DSSP) method [61].H-bonds between heavy atoms (*N* and *O*) as potential donors/acceptors were calculated with the following geometric criteria: (D⋯A ≤ 3.6 Å, pseudo-valent angle at H-atom ∠ D-H⋯A > 90°, where D (D = O/N) is a donor atom and A (A = O/N) is an acceptor atom). Hydrophobic contacts were considered for all hydrophobic residues with side chains carbon atoms within a 4 Å of each other.The radius of gyration (*R_g_*) was calculated from the atomic coordinates of the non-hydrogen atoms using the formula (Equation (1)) from [62]:
(1)$$R_{g}=\sqrt{\frac{\sum_{i=1}^{N}m_{i}r_{i}^{2}}{\sum_{i=1}^{N}m_{i}}}$$
where *m_i_* is the mass of atom *i*, *r_i_* is the distance of atom *i* from the center of mass of the protein.Clustering analysis was performed on the productive simulation time of each MD trajectory using an ensemble-based approach [27]. The analysis was performed every 100 ps. The algorithm extracts representative MD conformations from a trajectory by clustering the recorded snapshots according to their Cα-atom RMSDs. The procedure for each trajectory can be described as follows: (i) a reference structure is randomly chosen in the MD conformational ensemble, and all conformations within an arbitrary cutoff r are removed from the ensemble; this step is repeated until no conformation remains in the ensemble, providing a set of reference structures at a distance of at least r; (ii) the MD conformations are grouped into *n* reference clusters based on their RMSDs from each reference structure. The optimal cut-off was set to 4 Å for both clustered proteins or domains to allow comparison.To group the conformations with similar secondary structure patterns of each concatenated trajectory, the 8-letter alphabet defining secondary structure according to DSSP was simplified to a 5-letter alphabet where the secondary structure was assigned to a residue is in the ensemble {none; α-helix; 3_10_-helix; π-helix, strand}. The Jaccard distance was used to measure the pairwise dissimilarity between each conformation generated through MD simulation. Then, a complete linkage hierarchical clustering method [63] was performed for tree pruning distances from 0.05 to 1 each 0.05. At each clustering, the performance was assessed with the silhouette score [64]. The pruning distance leading to the best silhouette score clustering was further analysed.The Principal Components Analysis (PCA) modes were calculated for the backbone atoms (N, H, Cα, C, O) after least-square fitting on the average conformation calculated on the concatenated data. The eigenvectors were visualised with the NMWiz module for VMD [65].The cross-correlation matrices of the average conformation of each replicate were calculated with the R library bio3D [66] at 310 K with all available force fields.The relative Gibbs free energy of the canonical ensemble was computed as a function of two reaction coordinates (Equation (2) [67]):
(2)$$\Delta G_{\left(R1,R2\right)}=-k_{B}T\ln\frac{P_{\left(R1,\;R2\right)}}{P_{max}}$$
where *k_B_* represents the Boltzmann constant, *T* is the temperature. *P*_(*R*1,*R*2)_ denotes the probability of states along the two reaction coordinates, which is calculated using a k-nearest neighbour scheme and *P_max_* denotes the maximum probability.The conformational landscape was reconstructed on the concatenated trajectories of all proteins, the native and mutants, using the principal components (PC1 and PC2) obtained by the PCA after fitting on hVKORC1^WT^ TMD (t = 0).Pocket investigations were done with MDpocket with an isovalue of 0.5 on the concatenated data of each hVKORC1 species [26]. Well-defined grid points were exclusively extracted using VMD [68] and PyMol. Trajectory separation, with one conformation per 10 ps, was done with OpenBabel [64]. Conformational analysis was performed on nine conformations with a maximum P1 volume surpassing an arbitrary volume threshold, eight conformations with a maximum P2 volume surpassing an arbitrary volume threshold, and sixteen conformations characterised by minimal P1 and P2 pocket volumes.

#### Visualisation and Figure Preparation

Visual inspection of the conformations and figure preparation were performed with PyMOL (https://pymol.org/2/, accessed on 10 February 2022). To visualise the motions along the principal components, the Normal Mode Wizard (NMWiz) plugin [65], which is distributed with the VMD program, was used.

## Figures and Tables

**Figure 1 ijms-25-02043-f001:**
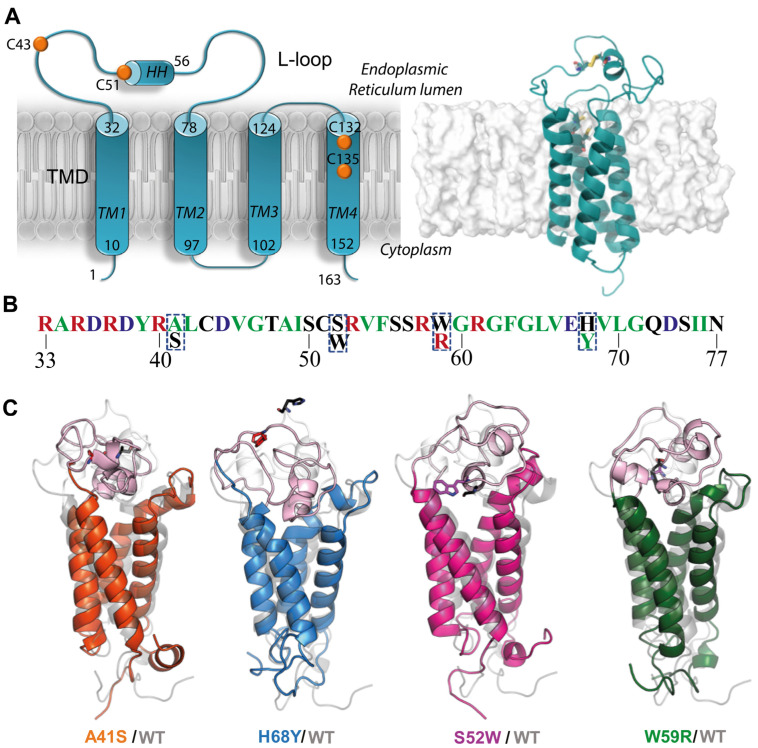
hVKORC1 and its mutants with missense mutations located in L-loop. (**A**) Four-helices topological model (left) and 3D de novo structure of hVKORC1 in reduced state (right) [19]. (**B**) L-loop sequence (R33-N77) with residues coloured according to their biophysical properties: positively and negatively charged residues in red and blue, respectively; hydrophobic residues in green; polar and amphipathic residues in black. Mutated residues are outlined with a dashed rectangle. (**C**) The 3D models of hVKORC1 mutants (in colour) superimposed with hVKORC1^WT^ (in grey). All models represent the last conformation of a randomly chosen MD trajectory (0.5 µs replicas 1–3). Proteins are displayed as cartoons with L-loop in pink and TMD in the colour specified for each mutant: hVKORC1^A41S^ in orange, hVKORC1^H68Y^ in blue, hVKORC1^S52W^ in magenta and hVKORC1^W59R^ in green. The native and mutated residues are drawn as black and coloured sticks, respectively. Labelling of structural domains (TMD, L-loop, N-, and C-terminals) and numbering of TMD helices (TM1-TM4) are shown in (**A**) on the left.

**Figure 2 ijms-25-02043-f002:**
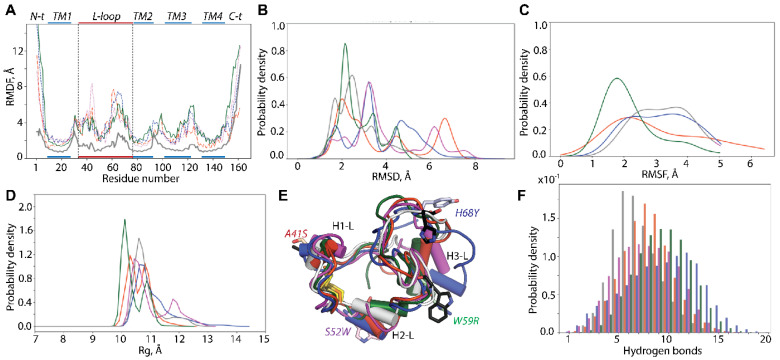
Conventional MD simulations of hVKORC1 mutants in the oxidised state. (**A**) RMSDs calculated for the concatenated data on the Cα atoms of each mutant and standardised to the initial coordinates of hVKORC1^WT^ (t = 0 µs). Probability densities of RMSD (**B**), RMSFs (**C**) and radius of gyration (Rg) (**D**). (**E**) Superposition of the most probable conformations (taken from the RMSD probability peaks) of each protein studied. (**F**) Number of hydrogen bonds stabilising the L-loop of each mutant compared to hVKORC1. (**A**–**F**) hVKORC1 mutants (with point mutations A41S, S52W, H68Y and W59R) are coloured in red, purple, blue, and green respectively; hVKORC1^WT^ is in grey.

**Figure 3 ijms-25-02043-f003:**
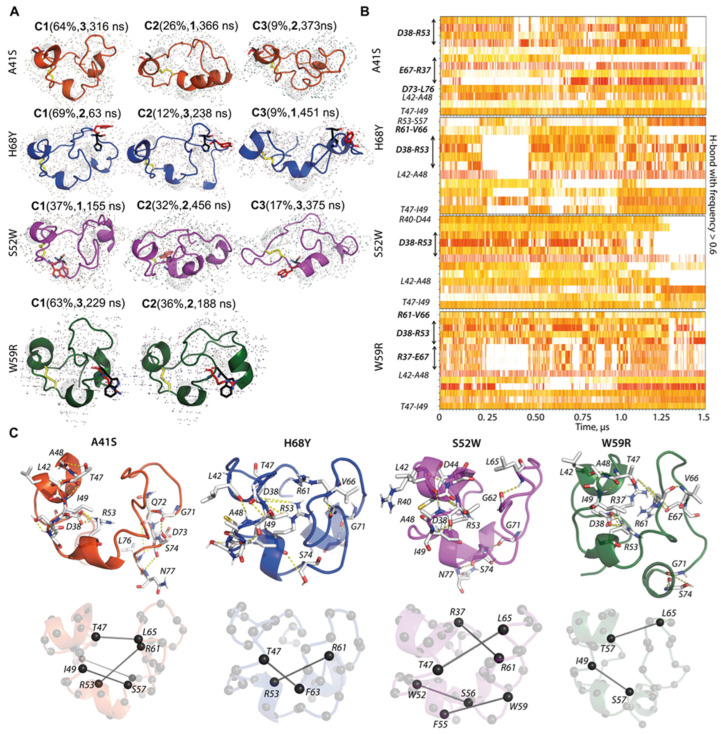
Characterisation of L-loop conformations. (**A**) Clustering of L-loop MD conformations for each hVKORC1 mutant. Clustering (cut-off value of 4.0 Å) was performed on each 100-ps frame of the merged trajectory of each mutant after fitting on its respective TMD domain. Representative L-loop conformations of clusters C^m^ (m = 1,2, …) with population ≥ 9% are shown. The population of each cluster is given in parentheses (in %), along with the replicate (numbered in bold) and the time (in ns) that the representative conformation was recorded. (**B**) Non-covalent contacts (time series of H-bond events observed with frequency ≥ 0.6) in each hVKORC1 mutant were calculated for the merged data (concatenated trajectories 1–3). The strength of contacts is represented by the colour: from the strongest (2.6 Å, in red) to the weakest (3.6 Å, in white). The helices stabilising H-bond interactions are considered but not labelled. (**C**) Non-covalent interaction patterns stabilising the predominant conformation in each mutant: (top) H-bonds (yellow dashed lines) and (bottom) van der Waals contacts (black lines) are shown on a representative conformation of the C1 cluster for each mutant. (**A**) The L-loop of mutants (in colour) is shown superposed with the L-loop of hVKORC1^WT^ (in grey). (**A**,**C**) Mutated proteins are distinguished by colour: hVKORC1^A41S^ (orange red), hVKORC1^H68Y^ (dark blue), hVKORC1^S52W^ (fuchsia), and hVKORC1^W59R^ (dark green). Disulphide bridges C43–C51 are drawn as yellow sticks, and mutated and native residues are shown in (**A**) as red and black sticks, respectively.

**Figure 4 ijms-25-02043-f004:**
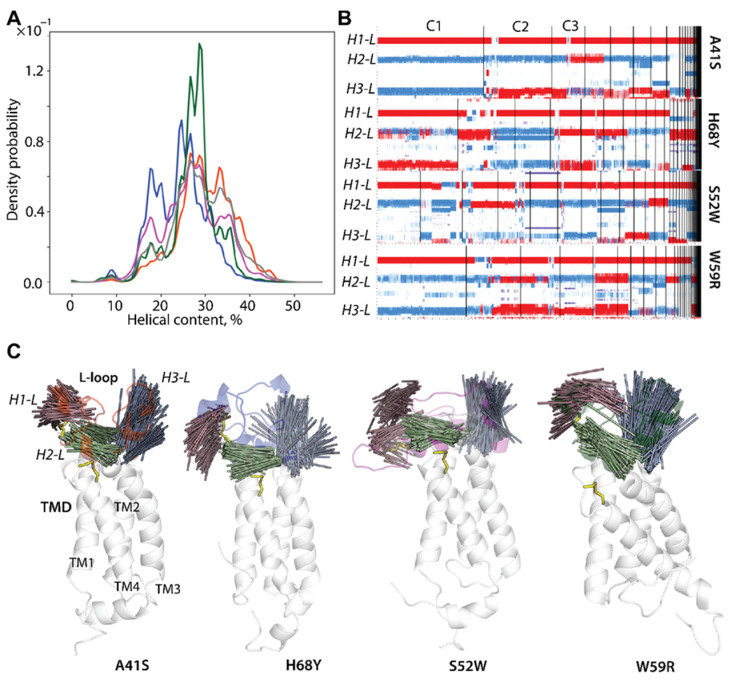
L-loop folding in hVKORC1 mutants. (**A**) Probability (in %) of helical content (α-, 3_10_-, and π-helices) estimated on the concatenated trajectories of each mutant. (**B**) Clustering of L-loop conformations using their secondary structure content. The α- and 3_10_-helices are in red and blue, respectively. (**C**) Drift of the L-loop helices observed over MD simulations of each mutant. The superimposed axes of the L-loop helices are projected on a randomly chosen conformation. The axis of each helix is defined as a line connecting two Cα atoms from the first and last residues from the helix. Helices H1-L, H2-L, and H3-L are shown in brown, green, and grey, respectively. Disulfide bridges are displayed as yellow sticks. Proteins (in cartoon) are distinguished by colour: hVKORC1^A41S^ in orange-red, hVKORC1^H68Y^ in dark blue, hVKORC1^S52W^ in fuschia, hVKORC1^W59R^ in dark green, and hVKORC1^WT^ in grey.

**Figure 5 ijms-25-02043-f005:**
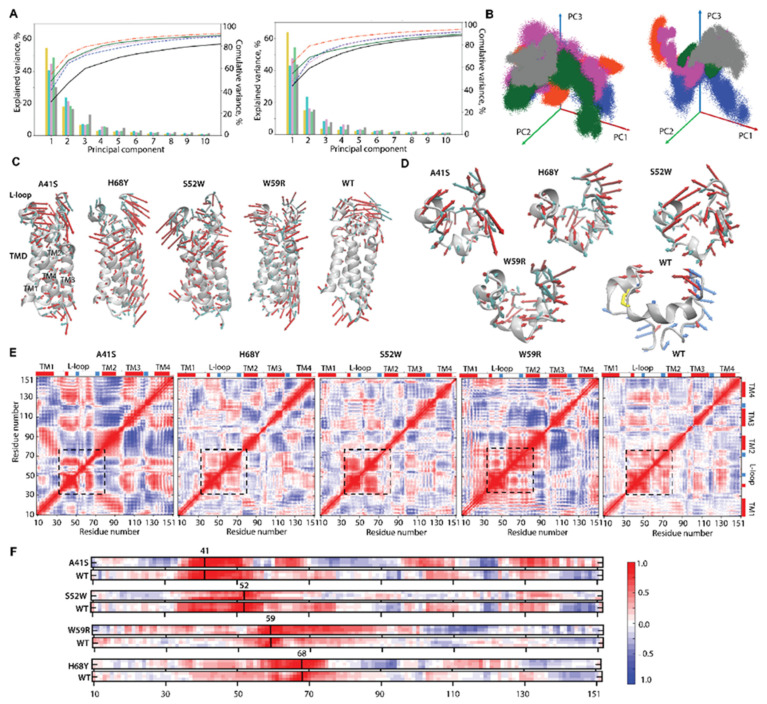
Inherent motion of hVKORC1 and its L-loop. (**A**) The PCA modes of hVKORC1, describing the TMD and L-loop (left), and only the L-loop (right) global motions, were calculated for the concatenated MD trajectory after least-square fit of the MD conformations to the initial conformation (t=0 µs) of the respective domain. Only the Cα-atoms were considered. The bar plot gives the individual explained variance (eigenvalue spectra) in descending order for the first 10 modes. Curves provide the cumulative explained variance. (**B**) Projection of MD conformations (the TMD and L-loop on the left and the L-loop on the right) in phase space along the first three principal eigenvectors. (**A**,**B**) Proteins are distinguished by colour: hVKORC1^A41S^ in orange-red, hVKORC1^H68Y^ in dark blue, hVKORC1^S52W^ in fuchsia, hVKORC1^W59R^ in dark green, and hVKORC1^WT^ in grey. (**C**,**D**) Atomic components in the first two PCA modes of the analysed hVKORC1 domains are drawn as red (1st mode) and cyan (2nd mode) arrows projected onto the respective average structure shown in a grey cartoon. Only movements with an amplitude ≥ 2Å are displayed. (**E**) The inter-residue cross-correlation map calculated for Cα atom pairs after fitting to the respective initial conformation (t = 0 µs) of each hVKORC1 using the concatenated trajectories. The L-loop region is delineated by a dashed square. (**F**) Correlation of each mutated (top) and native (bottom) residue with other hVKORC1 residues. (**E**,**F**) Correlated (positive, 1) and anti-correlated (negative, −1) motions between pairs of Cα atoms are represented by a red-blue gradient.

**Figure 6 ijms-25-02043-f006:**
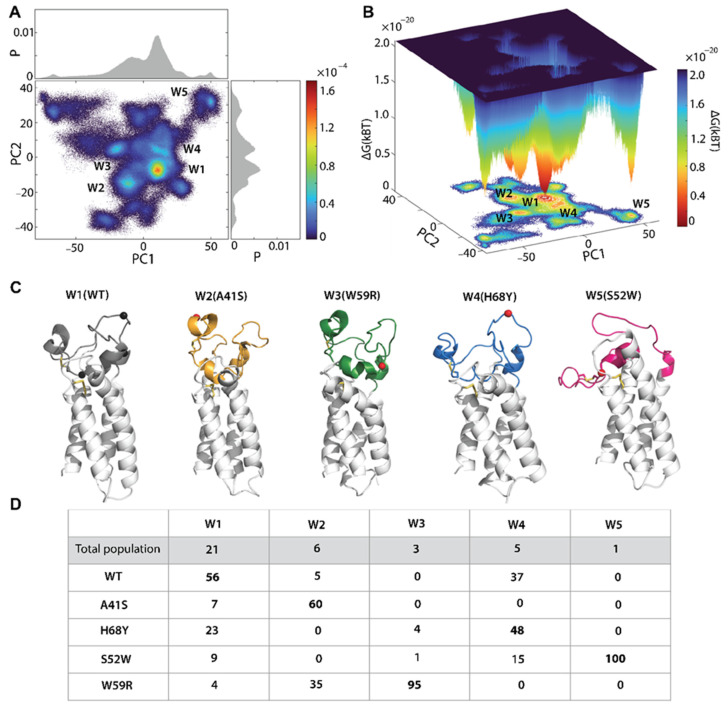
Free energy cumulative landscape (FECL) of hVKORC1^WT^ and its four mutants. The 2- (**A**) and 3-dimensional (**B**) FECLs are defined on PC1 and PC2 as the reaction coordinates. FECLs were generated on the 7.5 µs concatenated trajectory composed of MD conformations from all hVKORC1 proteins studied—hVKORC1^A41S^, hVKORC1^H68Y^, hVKORC1^S52W^, hVKORC1^W59R^, and hVKORC1^WT^. Blue colour represents the high energy state; green and yellow, low energy; and red represents the lowest stable state. (**C**) The most representative conformations taken at the minima of each well. Protein is shown as a cartoon with the L-loops distinguished by colour: hVKORC1^A41S^ in orange, hVKORC1^H68Y^ in blue, hVKORC1^S52W^ in fuchsia, hVKORC1^W59R^ in green, and hVKORC1^WT^ in grey. The position of the point mutation is shown as red (in mutants) and black (in hVKORC1^WT^) balls. (**D**). The content and its population (%) of each well (W1-W5) on the FECL. Assessment of the population for each hVKORC1 in a given well was estimated relatively to the total well population. The major population is in bold.

**Figure 7 ijms-25-02043-f007:**
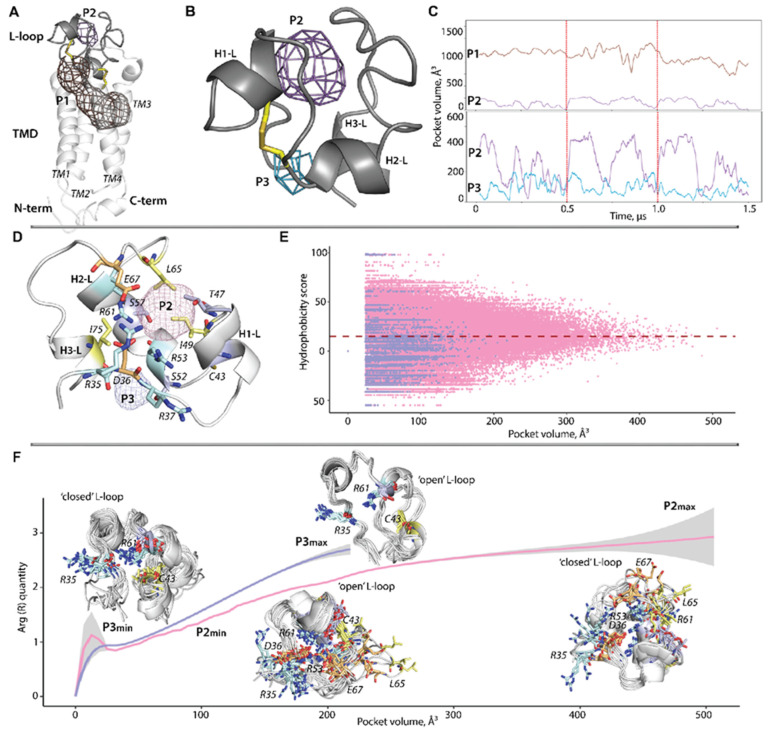
Pockets found in hVKORC1^WT^. (**A**,**B**) Two pockets localised in hVKORC1^WT^ and its L-loop. (**C**) Variations in volume in the pockets P1 and P2 (P3 is an upper part of the P1 pocket beginning at the L-loop). (**D**) Residues which form pockets P2 and P3. (**E**) Variations in pocket volume and hydrophilicity score. The dashed line shows the maximal value of the P2 volume. (**F**) Volume of pockets as a function of arginine quantity at the pocket surface. Conformations corresponding to the minimal and maximal values are shown together with the orientation of the side chains of residues forming the pockets surface. (**A**,**B**,**D**,**F**) Protein is shown as cartoon; disulfate bonds and crucial residues are in sticks; pockets are delimited by meshed contours.

**Figure 8 ijms-25-02043-f008:**
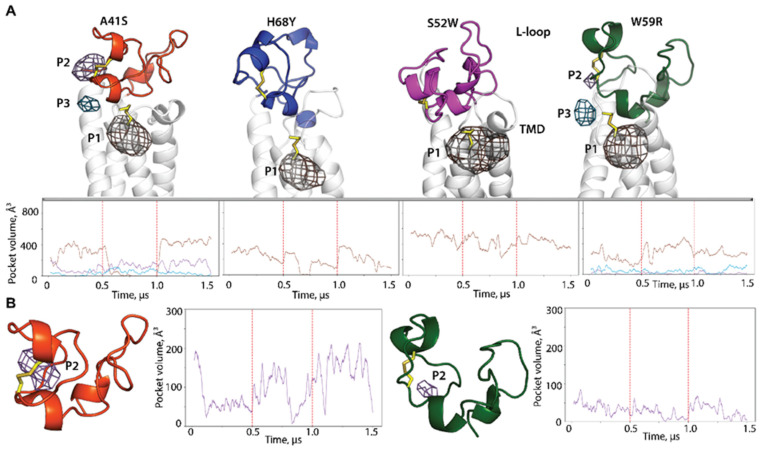
Pockets in hVKORC1 mutants. (**A**) Pockets localised in four mutants (top) and variation in their volume along the MD simulation (bottom). (**B**) Pocket P2 localised in L-loop and variation in its volume. (**A**,**B**) Protein is shown as a cartoon with the L-loop distinguished by colour: hVKORC1^A41S^ in orange-red, hVKORC1^H68Y^ in dark blue, hVKORC1^S52W^ in fuchsia, and hVKORC1^W59R^ in dark green. Disulfate bonds and crucial residues are in sticks; pockets are delimited by meshed contours.

**Figure 9 ijms-25-02043-f009:**
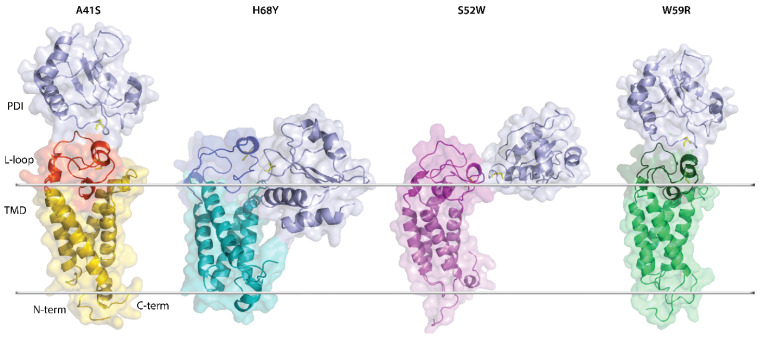
Computational protein–protein docking of PDI onto hVKORC1 performed with HADDOCK using an information-driven method. The best solution (most populated cluster C1) is shown for each hVKORC1 mutant. Proteins are presented as cartoons with PDI in lilac, and hVKORC1 mutants are distinguished by colour. In the PDI/ hVKORC1^A41S^ and PDI/hVKORC1^W59R^ models, the TMD and L-loop of hVKORC1 are denoted in yellow and red, and green and dark green, respectively. Disulfate bonds are in sticks. A possible boundary of the membrane is shown by grey lines.

## Data Availability

The numerical model simulations, upon which this study is based, are too large to archive or transfer. Instead, we provide all the information needed to replicate the simulations. The model coordinates are available from L. Tchertanov at ENS Paris-Saclay.

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
