# Peer review of "Synergy of Mutation-Induced Effects in Human Vitamin K Epoxide Reductase: Perspectives and Challenges for Allo-Network Modulator Design"

_ijms, 2024, doi:10.3390/ijms25042043_

Round 1
Reviewer 1 Report
Comments and Suggestions for Authors
The authors performed molecular dynamics simulations for studying the consequences of four L-loop mutations in the human VKORC1 protein. Based on the simulations, they discuss the conformational differences between mutations and analyze their effects on potential drug-binding pockets and the interaction with PDI. The manuscript is clearly written and the figures are of high quality.
I have the following recommendations for improving the manuscript:
1. The authors write that the hVKORC1 protein was simulated in an octahedral water box but they do not mention that the model systems contained a lipid bilayer. The authors should justify this as the lack of a membrane might have an effect on the conformation of the important regions in this transmembrane protein.
2. The authors compare the results for the hVKORC1 mutants to a simulation of the wild-type protein. However, such a simulation is not mentioned in the Methods section where the system preparation and the MD protocol is described. Was the same protocol applied to the WT system? The authors should clarify this.
3. The authors should discuss how the conformations of hVKORC1 mutants used for docking were obtained as this can affect the result of the docking.
4. In figure 3/B some of the analyzed non-covalent contacts are not labeled on the Y axis. The authors need to add the missing labels.
5. The caption of figure 8 is incomplete, the explanation is missing for part B.
6. The authors need to check the style of the reference list. For example, the number of each reference is usually written twice and the DOIs are web links for some references and not for others.
Author Response
Reviewer 1
Comments and Suggestions for Authors
The authors performed molecular dynamics simulations for studying the consequences of four L-loop mutations in the human VKORC1 protein. Based on the simulations, they discuss the conformational differences between mutations and analyze their effects on potential drug-binding pockets and the interaction with PDI. The manuscript is clearly written and the figures are of high quality.
Response: The Authors thank Reviewer 1 for the positive comments on the manuscript and critical remarks that were considered in the revised version of the manuscript or explained in the Rebuttal Letter.
I have the following recommendations for improving the manuscript:
- The authors write that the hVKORC1 protein was simulated in an octahedral water box but they do not mention that the model systems contained a lipid bilayer. The authors should justify this as the lack of a membrane might have an effect on the conformation of the important regions in this transmembrane protein.
Response: As we reported previously, the results (structural and dynamic metrics indicating the stable TMD and disordered L-loop) obtained by MD simulations of the de novo model of hVKORC1 inserted into a membrane (Chatron et al. 2017; Stolyarchuk et al., 2021) and placed in an aqueous solution (Ledoux et al., 2022) are identical. These results indicate that hVKORC1 is composed of the stable TMD and disordered L-loop. Furthermore, the de novo predicted model of hVKORC1 matches well with crystallographic structures (no membrane) (Liu et al. 2021; Ledoux et al., 2022). The most problematic part of hVKORC1 in these (and other) studies is the L-loop located in the endoplasmic reticulum lumen (solution). Based on these results, all proteins described in our manuscript were simulated in aqueous solution. However, since we observed that the missense mutations affect not only the L-loop but also part of the transmembrane helices, in perspective it will be interesting to analyse these mutants as inserted into the membrane to distinguish between the mutation-induced and membrane-stabilising effects. This text was added to the Discussions.
- The authors compare the results for the hVKORC1 mutants to a simulation of the wild-type protein. However, such a simulation is not mentioned in the Methods section where the system preparation and the MD protocol is described. Was the same protocol applied to the WT system? The authors should clarify this.
Response: The wild-type hVKORC1 was described in our previous papers (Chatron et al. 2017; Stolyarchuk et al., 2021; Ledoux et al., 2022) together with the system preparation and MD protocol. The hVKORC1 mutants were studied using the same protocol, and generated data were analysed by the same methods with using the similar criteria as for the native protein. In our present paper, for the mutants’ and wild-type comparison, we used either the early reported characteristics for the native hVKORC1 or the novel metrics obtained by using the previously generated datasets. This aspect was added to the Results.
- The authors should discuss how the conformations of hVKORC1 mutants used for docking were obtained as this can affect the result of the docking.
Response: Similar to the docking trials for native hVKORC1, for docking of mutants we used two different conformations with L-loop shapes (i) compact (closed, most probable) and (ii) elongated (open, least probable). The results were comparable, but the clusters’ populations were greater for hVKORC1 with an L-loop in a closed conformation, and the docking solutions were more interpretable, with PDI in the lumen and not in the membrane. These results were presented in the manuscripts.
- In figure 3/B some of the analyzed non-covalent contacts are not labeled on the Y axis. The authors need to add the missing labels.
Response: The Y axis was labeled.
- The caption of figure 8 is incomplete, the explanation is missing for part B.
Response: The caption was completed.
- The authors need to check the style of the reference list. For example, the number of each reference is usually written twice and the DOIs are web links for some references and not for others.
Response: The original submitted file contains only the single enumeration of the references (see Manuscript in pdf format). All references were generated by using EndNote X8. I am not sure that converting “active” citations into plain text, allowing the bibliography to be edited, is desirable for the editor. Usually I send both versions (original and revised) with "active" citations, and only in Proofs I complete the required edition.

Reviewer 2 Report
Comments and Suggestions for Authors
The authors investigated the synergy of mutation-induced effects in Human Vitamin K2 Epoxide Reductase. The work addresses an intriguing topic within the scientific community. The article is well-crafted, excellently written, and the scientific content is robust. From my perspective, the article can be accepted after minor corrections.
-The article's theme is crucial for the scientific community. However, in the introduction, out of the 36 cited articles, only 4 are recent (2022/2023). The authors should seek more current references to showcase the latest developments in this field.
-The authors employed the MODELLER homology modeling approach. Why did the authors not explore the use of AI tools to compare modeling results? Integrating AI-based tools could enhance the depth of the analysis and provide additional insights into the modeled structures.
-The authors chose the SHAKE algorithm for their study. While effective, SHAKE can impose a substantial computational burden. Could the authors provide rationale for selecting SHAKE over LINCS? A brief discussion on the computational considerations behind this choice would be valuable.
-The authors explored various physicochemical parameters influencing protein structure. However, the analysis did not include an evaluation of the distribution of electric charges in the protein. Electric charges play a fundamental role in stabilizing different protein conformations. The authors should calculate the Molecular Electrostatic Potential (MEP) for both the wild-type and mutant proteins and incorporate a discussion on charges in their work.
Comments on the Quality of English LanguageMinor editing of English language required
Author Response
Reviewer 2
Comments and Suggestions for Authors
The authors investigated the synergy of mutation-induced effects in Human Vitamin K2 Epoxide Reductase. The work addresses an intriguing topic within the scientific community. The article is well-crafted, excellently written, and the scientific content is robust. From my perspective, the article can be accepted after minor corrections.
Response: The Authors thank Reviewer 2 for the positive comments on the manuscript and critical remarks that were considered in the revised version of the manuscript or explained in the Rebuttal Letter.
-The article's theme is crucial for the scientific community. However, in the introduction, out of the 36 cited articles, only 4 are recent (2022/2023). The authors should seek more current references to showcase the latest developments in this field.
Response: The recent papers were added. I note that the PubMed searchfor the recent publications (last 5 years) corresponding to the topic of interest, ‘human VKOR mutant, resulted in a very poor number of articles.
-The authors employed the MODELLER homology modeling approach. Why did the authors not explore the use of AI tools to compare modeling results? Integrating AI-based tools could enhance the depth of the analysis and provide additional insights into the modeled structures.
Response: Each modelling approach produces only an approximate model. We used the Modeller to produce such approximate models. As each model was further studied by MD simulation (3 replicas, each of 0.5 µs), it is a perfect process for the model refinement.
-The authors chose the SHAKE algorithm for their study. While effective, SHAKE can impose a substantial computational burden. Could the authors provide rationale for selecting SHAKE over LINCS? A brief discussion on the computational considerations behind this choice would be valuable.
Response: As we studied the mutation-induced effects, we compared all mutant-related data with the native protein used as a reference. To be able to do such a comparison, we used the same protocol (and the SHAKE algorithm) for the data generation and their analysis as for the native hVKORC1, described in our previous papers (Chatron et al. 2017; Stolyarchuk et al., 2021; Ledoux et al., 2022). The choice of the SHAKE was explained in these papers.
-The authors explored various physicochemical parameters influencing protein structure. However, the analysis did not include an evaluation of the distribution of electric charges in the protein. Electric charges play a fundamental role in stabilizing different protein conformations. The authors should calculate the Molecular Electrostatic Potential (MEP) for both the wild-type and mutant proteins and incorporate a discussion on charges in their work.
Response: The estimation of the Molecular Electrostatic Potential (MEP) that we plan to carry out after a study of each protein and its complex by MD or QM/MM simulations, similar to that performed on ketosteroid isomerase (Wang et al. 2021). Such a study is the subject of another article.
